# Android Spyware Detection Using Machine Learning: A Novel Dataset

**DOI:** 10.3390/s22155765

**Published:** 2022-08-02

**Authors:** Majdi K. Qabalin, Muawya Naser, Mouhammd Alkasassbeh

**Affiliations:** Department of Computer Science, Princess Sumaya University for Technology, Amman 11941, Jordan; m.aldalaien@psut.edu.jo (M.N.); m.alkasassbeh@psut.edu.jo (M.A.)

**Keywords:** spyware, spying systems, stalkerware, machine learning, random forest, privacy, spyware dataset

## Abstract

Smartphones are an essential part of all aspects of our lives. Socially, politically, and commercially, there is almost complete reliance on smartphones as a communication tool, a source of information, and for entertainment. Rapid developments in the world of information and cyber security have necessitated close attention to the privacy and protection of smartphone data. Spyware detection systems have recently been developed as a promising and encouraging solution for smartphone users’ privacy protection. The Android operating system is the most widely used worldwide, making it a significant target for many parties interested in targeting smartphone users’ privacy. This paper introduces a novel dataset collected in a realistic environment, obtained through a novel data collection methodology based on a unified activity list. The data are divided into three main classes: the first class represents normal smartphone traffic; the second class represents traffic data for the spyware installation process; finally, the third class represents spyware operation traffic data. The random forest classification algorithm was adopted to validate this dataset and the proposed model. Two methodologies were adopted for data classification: binary-class and multi-class classification. Good results were achieved in terms of accuracy. The overall average accuracy was 79% for the binary-class classification, and 77% for the multi-class classification. In the multi-class approach, the detection accuracy for spyware systems (UMobix, TheWiSPY, MobileSPY, FlexiSPY, and mSPY) was 90%, 83.7%, 69.3%, 69.2%, and 73.4%, respectively; in binary-class classification, the detection accuracy for spyware systems (UMobix, TheWiSPY, MobileSPY, FlexiSPY, and mSPY) was 93.9%, 85.63%, 71%, 72.3%, and 75.96%; respectively.

## 1. Introduction

Spyware detection is the process of identifying programs that collect data for the purpose of espionage activity [1]. Smartphones are becoming an essential requirement of daily life. By 2023, it is expected that there will be four billion people using smartphones. The Android operating system is the most extensively utilized in the mobile devices market. In May 2021, it had a 70% share of the market. Apple iOS has 26.99 percent of the market share; other smaller suppliers make up the remaining 3 percent [2]. The official applications store for Android smartphones is Google Play. As of May 2021, there were more than 2.9 million applications on it. Of these, AppBrain categorizes 2.5 million as “normal applications” and the remaining 0.4 million as “low-quality apps”. Viruses and malware are more likely to attack Android systems because of their extensive distribution, making it an easier target for thieves. Various strategies for identifying these attacks have been presented, and machine learning is among the most notable. This is because machine learning algorithms may generate a classifier from a complex collection of instances. By using examples instead of defining signatures directly, spyware detectors might avoid having to write more code. It is a difficult and time-consuming process to come up with signature definitions for all possible attack situations, and although there may be no clear rules (signatures) for some of them, examples may be found very simply, which is the main concept in machine learning [3].

Since the first android smartphones were introduced with Android 1.6 Donut, Android’s internal security has significantly improved. Since the introduction of Google Play Protect, the privileges of applications have been substantially reduced since they now have to directly obtain all permissions from the user. Security has also been relocated to a separate, updatable component that is not tied to a specific vendor [4].

However, the ability to install applications (apps) from untrusted sites is a security vulnerability that exists in recent Android versions. This is a true “window of opportunity” for hackers. For this reason, a slew of third-party platforms for the distribution of Android applications has appeared. Downloads from these sites vary from famous software clones to various spyware classes. In addition to the platform, however, there are other dangers. Apps may also be loaded and installed into the system using the client that works with it, similar to the official “Google Play” client. Allowing the installation of these applications actually provides a wide scope for violating the privacy of the smartphone, and the numbers confirm this, according to a report by Kaspersky [5]. In the past two years, attacks against Android phones have increased dramatically, as shown in Figure 1.

One of the most common complications that causes scientific confusion is the failure to clarify the exact difference between malware and spyware. In terms of technical operations, the technical details of malware differ from spyware, even if spyware is a subclass of malware [6]. Therefore, it is impossible to rely on malware datasets to propose detection models through machine learning capable of detecting commercial or non-commercial spyware systems. According to Kaspersky, spyware is informally described as harmful software meant to penetrate your computer or mobile device, capture data about you, and transfer it to a third party without your knowledge [7]. In today’s world, spyware is one of the most prevalent dangers to organizations and individuals alike on the internet since it can steal and harm vital data; personal information is collected and sent to advertising, data businesses, or other external users through spyware [8].

The term malware refers to software installed without the user’s permission with the express purpose of causing harm to or destroying systems. Malware is any malicious program that is meant to damage your computer. Cybercriminals use malware to obtain illegal access to a system, destroy your data, or even lock your machine. Malware often functions unnoticed, and you will not realize that your machine is infected until after the damage has been done. Thus, the word malware is used to describe a wide variety of malicious software, such as viruses, ransomware, spyware, trojans, adware, stalkerware, and scareware) [9].

It important to explain methodologies used in spyware detection models in term of analysis, detection, and deployment. Spyware is purposely created to spy on the mobile device without the smartphone user’s knowledge. One of the main problems in spyware detection systems is that they do not actually depend on analyzing the behavior of applications and only rely on superficial information related to the digital signature of these applications [10].

Some approaches have been developed to assist researchers in detecting and overcoming the existence of spyware. Therefore, before addressing these studies, we must talk about the basic architecture on which those studies were built. Figure 2 describes the various spyware detection approaches [11]. 

Analysis techniques can be classified under two approaches, dynamic and static. The analysis technique represents the way in which the detection system handles the analysis phase. Static and dynamic approaches will be explained in detail [12]. Detection approaches, in general, can be classified under three classes: anomaly, which depends on the behavior of the spyware; a signature-based class, which depends on spyware milestones; and finally—the most successful model—the hybrid model, which adopts both anomaly and signature. For the deployment, there are three cases in which the detection approaches can be used: host-based, in which the detection system works on the target device; network-based, which depends on the processing of data through the network infrastructure; and hybrid-based, which adopts both host-based and network-based with strict conditions [13].

AI includes a subfield called “machine learning” that aims to mimic the learning process of humans by using data and improve its accuracy over time. Analytical model construction is automated via the use of machine learning. In this field of AI, computers are trained to learn from data, recognize patterns, and make choices on their own with the lower amount of human input possible. Data science is becoming more dependent on machine learning. In data mining projects, algorithms may be taught to generate classifications and predictions using statistical approaches [14]. 

This study aims to present a novel dataset for the most advanced Android-based spywares that are commercially available. In addition, it presents a proposed detection model based on that dataset. Therefore, the objectives of this work can be summarized as follows:First, provide a profound understanding of the concept of spyware and common headlines about spyware detection approaches.Second, discuss literature review that proposed Android-based spyware detection model approaches. To understand the knowledge gap and determine the weakness of each of these studies.Third, create a novel benchmark dataset based on a unified activity list for the top five advanced spyware systems (UMobix, TheWiSPY, MobileSPY, FlexiSPY, and mSPY). The dataset includes network traffic data collected in two phases: the installation phase and the operation phase.Fourth, evaluating and discussing the experimental results of the proposed detection model regarding the F-measure, true positive rate, and accuracy. A proposed model is conducted based on a novel dataset.Ultimately, validate the novel dataset legibility using a confusion matrix analysis.

The main contribution of this paper is the dataset itself, along with the dataset collection methodology; further, a detection model for Android-based spyware is proposed. 

## 2. Background and Related Works

Because of the lack of technical information and the difficulty of obtaining technical reports about advanced spyware systems for smartphones, there are a limited number of studies that mention the advanced spyware systems targeted in this paper, and a large part of the research confuses malware and spyware. Therefore, a literature review of Android spyware is a critical analysis of the extant literature on a topic. Such a review aims to synthesize what has been learned from past research and provide insight into future research. Four common approaches to a structure literature review are chronological, thematic, methodological, and theoretical. In our case, the thematic structure is the most optimal. We have arranged our data depending on the relevance and significance of the many literature sources we have gathered. 

The FlexiSPY, Mobilespy, mSPY, TheWiSPY, and UMobix spyware tools are the spyware apps targeted explicitly in this study and any other commercial spyware tool mentioned belongs to the same classification. The following methodology was adopted to select research papers related to the literature review: A title keyword-based search on “Android” and “Spyware” was conducted, along with a content keyword-based search on “FlexiSPY, Mobilespy, mSPY, TheWiSPY, and UMobix”.

Using network traffic analysis, Conti et al. [15] proposed what can be considered the first Android-based detection model based on network traffic. The rooting approach provides applications with extensive authority over the mobile phone. This is one of the most significant technical flaws in this study regarding the data collection phase, as this research does collect network traffic from Android devices without root permissions. Depending on whether the phone is rooted or not, the network data collected by a spying system will differ. Therefore, the data collected cannot be relied on. As for the iOS phones, the research does not mention technical limitations that restrict mSPY from accessing iCloud data only. This study did not address other applications from the same class and only focused on “mSPY”. Additionally, it is essential to note that the study did not address the issue of the mSPY program’s accessibility to social networking applications, such as WhatsApp, where numerous technical details are reported and not included in the study [16].

Using Aspect droid, Ali-Gombe et al. [17] developed an application-level approach to investigate Android apps for potential malicious behavior. A study proposes spying on Android applications to detect if they are adopting spying activities. This study does not include any datasets and is limited to application-level scenarios.

Saad et al. [18] proposed monitoring the permissions requested by the application and classifying the applications as malicious and healthy accordingly. This study did not provide real solutions for collecting and analyzing network traffic data and relied on a simple and basic way of analyzing the required permissions. The significant gap in this research is that spyware applications work in the case of rooting. In this case, the permissions granted to the application cannot be known. Thus, it cannot be relied upon to detect the mentioned spying apps.

Carlsson et al. [19] describe an app called KAUDroid that monitors smartphone permission requests. A web interface provides access to information about user permissions. This was an attempt to analyze the permissions of the applications rather than provide any real meaning for detecting spying tools. As we have said before, the permissions granted to applications can no longer be trusted with the new spyware mechanism of action in place.

The research of Abualola et al. [20] addressed the spying problem by following up and monitoring notifications. In order to gain access to incoming messages, spy systems rely on notifications. Encryption prevents them from being retrieved from the phone’s internal application databases. Advanced spyware exploits the notification to obtain sensitive messages because the messages remain locally encrypted on the phone. Researchers tested the Galaxy S4 with two different Android versions, 4.3 and 5.0, to see if any malware was present. Several popular apps were tested for their ability to spy on other people’s conversations. It was found that, WhatsApp and Facebook Messenger notifications content could only be taken from Android devices running Android 4.3 or lower. In both Android 4.3 and Android 5.0, however, BBM and SMS messages can be sent to the attacker.

The primary goal of Pierazzi et al. [21]’s research was to develop a model that characterizes spyware. In their research, malware and spyware were relied upon without distinguishing between them. As clearly stated in this research, researchers relied on the following malware: RACE-CARD, HEHE, PINCER UAPUSH, and USBCLEAVER. However, the stated malware tools are entirely different from the spyware applications that spy on Android smartphones and cannot be relied upon for building a spyware detection model. For example, they can only collect limited information about social media services such as Facebook or bank accounts, and their functions are minimal. Using VirusTotal, they have downloaded about 5000 spyware tools. Most of the tools that have been reported for VirusTotal service include many samples between malware and spyware, and they do not include spy tools or complete spying systems. As a result, even if attributes are extracted uniquely, there will be no actual accurate data.

Han et al. [22] proposed a model to detect some of the most common attacks related to malware, the FARM (Feature transformation-based AndRoid Malware) detector. It is important to note that they have significantly contributed to the assembly of three new and diverse classes of feature transformations that permanently transform Android malware prediction’s original feature space. There are three types of transformations: landmark-based, feature-clustering-based, and correlation graph-based. In addition, numerous tests combining the new characteristics with those of standard detectors and a post-fusion phase were assumed as the second contribution. For that study, they tested the FARM strategy with six different types of Android malware: rooting applications vs. goodware, rooting applications vs. other malware, spyware vs. goodware, spyware vs. the other malware, banking trojans vs. goodware, and other malware vs. banking trojans. FARM’s efficacy is tested in a total of 12 different studies.

Compared to other baseline systems, FARM is nearly twice as resistant to these three threats as the baseline systems. Furthermore, FARM discovered two rooting programs in Android APKs before any of the 61 antiviruses on VirusTotal had done so. The data relied upon in this study had nothing to do with spyware apps for Android phones [23].

Kaur et al. [24] proposed a detection framework based on description analysis, permission mapping, and interface analysis. The researchers proposed gathering data from three primary sources, description mapping, interface layout, and source code analysis, along with all permissions gleaned from the XML file; thus, these data are used to create pre-digital signatures for applications that may engage in espionage activities. This research suggests relying on a broadcast receiver that issues an alert whenever a new or updated application is installed or updated in the system. After receiving an alert, the recipient locates the .apk file and uploads it to the server, even though the Play Store has hidden the .apk file by default. A real-time server is required for this approach. Various programs are scanned for malware using antivirus software, such as McAfee, Avira, and others.

In research by Vanjire et al. [25], digital samples or system calls were reviewed to understand the malicious application’s runtime behavior. Because of the wide variety of malware families and execution contexts, the system’s built-in malware detection and threat classification capabilities can be used to identify threats. In addition, supervised machine learning methods are used to implement various threat alert strategies. They make use of a behavioral report and a machine-learning algorithm. Subsequently, MDTA was put through its paces in various datasets and environments. Data on spyware-related comprehensive surveillance systems were missing from this study.

According to Sutter et al. [26], this research analyzed the background services on the Android operating system and the activity of several applications that engage in spying activity. The main problem in this research is that it does not depend on the data in actual spyware applications such as those targeted in this research.

Malik et al. [27] developed a method called CREDROID, which can detect malicious programs based on their DNS requests and the data they communicate to distant servers. It is impossible to rely on the DNS requests to detect spying tools, as most advanced spying systems periodically change their internet address and even the DNS records.

Anshul et al. [28] proposed a network traffic analyzer for malicious activities. This detector will catch malware that operates on Android devices by taking advantage of background connections. A problem that affects both Android and iOS users has to be addressed.

Taylor et al. [29] suggested a detection model based on data from typical applications; applications not classified as spying tools have been relied upon. Even if they transfer data from the user’s device to an external server, it is all done with the user’s knowledge.

After conducting the previously explained methodology and reviewing mentioned references, we concluded that: No dataset was found for such commercial spyware systems;No verifiable network-based detection model is available for such tools;Most of the research reviewed adopts permission-based parameters with no reliable model.

A complete analysis of the searches that were found according to the keyword search mechanism that was adopted has been conducted. A summary of the reviewed state-of-art is listed in Table 1.

## 3. Dataset

This section will be dedicated to discussing the dataset in terms of identification, the mechanism of data collection, data preparation techniques, data structure, composition, and finally to benchmark data. All dataset files will be shown in a detailed table, knowing that dataset contents are available for public use for research purposes under CC BY 4.0 license. 

### 3.1. Dataset Identification

This section will list the identification details about the dataset, title, data type, data class, data source, targeted applications, data format, and other identification parameters needed for the researchers. The dataset includes network traffic collected from five targeted spyware applications used commercially. The hybrid deployment approach was conducted within the process of data collection. Each spyware application was activated within its full features, and the smartphone used for data collection was rooted using “Dr. Fone Root”.

This dataset does not include any private data that violate the privacy of any person. The data collection process considers all relevant laws and conforms with the legislation listed within GDPR and PDPC regulations. The published version of this dataset will include a written acknowledgment from the person(s) whose data may be contained in this dataset that they have no objection to publishing any data that may lead to them in the future. All dataset main identification parameters are listed in Table 2. The dataset content is explained in detail within the next section.

### 3.2. Dataset Structure

In this section, we will list and explain the content of the novel dataset proposed in this paper. We will list all the files within the dataset along with file details and hashes. This dataset includes a total of 24 files of 350 MB. This dataset includes network traffic data collected using a packet sniffer tool operated over an Android-based smartphone under pre-defined rules and conditions. Spyware systems selected within this dataset are listed in Table 3, with the relevant commercial and compatibility details.

It was necessary to study these applications at the level of their internal design and analyze the packages of these applications. For this purpose, the Apktool system was used. Apktool is a reverse engineering tool identifying the programming platforms used to develop these apps [30]. Regarding reverse engineering, our interest was limited to identifying the programming platforms. Each spyware system scope, the platform used for development, upload mechanisms, and sniffing strategy, is presented in Table 4. Spying scope explains the scope of the spyware system as each spyware provides different spying scope, representing the main point of advantage for each spyware. The platform represents the development platform used to program that spyware; all of these spyware systems as presented were based on Java. The upload strategy explains the time constraint mechanism in which each spyware uploads data to the C2C server [31]. Sniffing is one of the essential internal features that must be considered in analyzing network traffic behavior. We identified each sniffing strategy according to the spyware documentation and used the previously mentioned Apktool for reverse engineering.

The dataset list of files along with basic identification info regarding dataset metadata, including file name, file size, MD5 Hash, and DataTag, are listed in Table 5. MD5 is listed to protect the file’s integrity.

Figure 3 shows the data volume distribution. Each system’s data volume differs from the others, even though they have the same period and constraint. The different behavior of these applications can explain the variation in the size of the data. The behavior of these applications differed in the network activity, which is directly related to the mechanism of developing these applications and the mechanism they use to send data; however, it is not one of the objectives of this study.

A total of 386,963 packets were collected within the data sets. CSV files can provide a high-level understanding of data distribution in any dataset. Thus, we have analyzed all CSV files as a high-level data representation to analyze network traffic based on protocol type. Table 6 shows the data distribution for each protocol within all dataset classes.

As shown in Figure 4, the data represents the protocol type statistics for all files in the dataset. The TCP protocol is the most significant since it is the primary protocol for communication and data transfer between the various applications and their C2C servers. Table 6 shows a complete statistics list for each protocol in terms of class. As explained before, Class A represents regular traffic, Class B represents spyware installation traffic, and Class C represents spyware traffic under operation.

### 3.3. Data Collection Methodology

A novel approach has been adopted to collect data to maintain six main factors: quality of data volume, quality of data format, quality of data diversity, quality of data rationality, quality of data eligibility for use in machine learning, and quality of data documentation. A networking technique known as packet capture involves intercepting data packets within OSI layers. Data link, network, transport, session, presentation, and application layers data were collected and packed in this experiment. The data collection methodology was divided into three main sections according to the data class: Class A, Class B, and Class C.

The sniffing process continually monitors and captures all data packets traversing a network or device. Network administrators use sniffers to monitor and troubleshoot network traffic, and researchers use them to build detection models by analyzing sniffed data using machine learning. Sniffers are implemented in the system as either hardware or software. 

A hybrid deployment approach was adopted in the sniffing process, which includes collecting network traffic data at the packet level using a PCAPdroid sniffer locally from the smartphone. Transmission control protocol (TCP), user datagram protocol (UDP), and internet control message protocol (ICMP), among others, underpin packet-based data. 

As mentioned before, the dataset collection methodology adopted three classes of data, Class A, Class B, and Class C, listed as follows:Class A: This class represents smartphone standard data traffic needed during the classification process using machine learning. This class collected data without any active spyware within the smartphone. This means data collected in this class represent clean data. It does not include any malicious activity of any application on the phone, which was ensured by performing a complete format process of the phone before collecting this data.Class B: This class represents network traffic during the spyware installation. These data can be used to develop a pre-infection detection model. In this stage, we collected data that can be used in hybrid approach detection: spyware package file traffic and spyware exchanged packets between C2C and infected devices. Briefly, this stage represents the data collected during the downloading and installing stage of the spyware.Class C: This class represents standard spyware traffic for each spyware system. In this class, data were collected by operating a spyware system with full features. In this class, a unified list of activity applied over the smartphone to guarantee data consistency within all data collected from all other spyware. These data can be used to develop a post-infection detection model.

A unified activity list is a pre-set list of activities sequentially. The activity list includes specific events and activities applied within specific time constraints to have identical experimental conditions. Activity list events and actions are listed within Table 7.

#### 3.3.1. Class A—Collection Methodology

As shown in Figure 5, the process starts with a mobile format that guarantees that no other apps are installed; then, PCAPdroid is installed along with other social apps listed within the activity list. To use social apps, we must activate them on an actual phone number and complete the verification process. After checking the installation process for PCAPdroid and other installed apps, we activate PCAPdroid and other social apps according to the listed activity list. This step is meant to collect clean network traffic without any malicious traffic [32].

#### 3.3.2. Class B—Collection Methodology

As shown in Figure 6, Class B represents spyware installation traffic data. Mobile formatting is critical in each phase. As shown, installation and downloading start after running the PCADPdroid app. Once we downloaded the spyware package, we collected all packets content that can be used for packet-level detection and can provide vast data about spyware digital signatures. At this stage, we are collecting the necessary signatures without encryption because we collect the packet information up to the seventh layer of the OSI layer. This class of data is used for the post infection process.

All spyware systems targeted in this study have the identical installation mechanism in terms of procedures, and these procedures were followed accurately as described by the manufacturers of these systems [33]. The network traffic collection process must include spyware package data and operational data. Here, it is essential to explain the mechanism for installing these applications, followed during the data collection process. Five essential steps to install these systems are explained in detail as follows:1—Purchase spyware subscription: Each spyware system has specific features and prices. Most of the vendors provide almost the same features with a tiny difference between each one of them.2—Disable security notification on Play Store: Play Store provides a security mechanism that alerts the user whenever a malicious application is about to be installed. This step is critical and guarantees hiding security notifications.3—Downloading spyware package: In this step, the user downloads a spyware system package; spyware providers usually use anonymous servers as a source for the package downloading process.4—Installing spyware package: The installation process is fast and easy; you have to follow the instructions and have your system running. Each system uses a code name the user can identify within the Android apps list. The spyware system’s names are listed in Table 8.5—Activate monitoring panel: The control panel provides access to various data types.

#### 3.3.3. Class C—Collection Methodology

Class C represents the data for the spyware system while it operates on the smartphone. That is, while carrying out its actual espionage activity. From here the actual value of the unified activity list is obtained, which is applied while collecting the data in this class in a clear, predefined order. As shown in Figure 7, the workflow and after starting PCAP droid. It is essential to apply a unified activity list accurately to ensure the integrity of the data between the various applications and achieve similar measurable conditions in the analysis stage.

### 3.4. Dataset Benchmark

Producing high-quality benchmark datasets is a complex and time-consuming operation—one of the most critical phases in dataset preparation. A dataset to be accurately labeled as a benchmark must first be evaluated according to predetermined criteria benchmark. Objectively interpreted, comparable, and repeatable algorithm benchmarks are essential to validate any dataset. In general, benchmarks are more helpful and essential if they can be shown to be beneficial. However, the benchmark dataset does not have to be exhaustive or conclusive to be suitable for the job. Dataset copyrights must be available to the public. This signifies that it is licensed under a permissive and open license [34].

A benchmark dataset has to be accessible and available to the general public. For example, everyone should have access to it without enrolling on a webpage or waiting for an email, without ever doing anything that halts their research rate. The openness of a dataset should ensure that it can be copied elsewhere. It must have sufficient characteristics to be engaging. It should be the result of genuine experimental activity. The characteristics should be independent; derivatives may be generated at any time. Therefore, this section will list a table representing the most reliable criteria for scientifically measuring the benchmark. A benchmark is a predetermined metric for assessing the quality of a product or service. Variance is the difference between the benchmark and the statistic against which it is compared [35]. The deviations from the stated benchmark are used to determine whether or not a data rule has continuously met or exceeded its objective for data monitoring. Benchmarks may be used for any statistics arising from data rule testing or metric value calculations—Table 9 lists our novel dataset’s main benchmark features.

## 4. Proposed Model

In order to determine if a dataset is eligible for use in a detection model, it must be tested and analyzed. The model results are an actual and scientific examination of the dataset, especially those collected in real environments without using virtual data generation tools. In this section, we will explain our proposed model, a model designed to detect Android-based spyware systems in two scenarios: pre-infection and post-infection.

### 4.1. Baseline Algorithm

The random forest algorithm is a supervised learning algorithm for classification and regression. It creates a random forest, or ensemble, of decision tree nodes. This improves the accuracy of predictions to create an effective model with low bias and high variance. This algorithm is practical when many variables are present in the dataset. The goal is to predict the value derived from these variables based on multiple inputs. Random forests are often used as predictive analytics in data mining or applied statistics problems to identify patterns in large data sets [36]. It uses a voting process at prediction time to obtain the classification or regression votes from individual trees. The algorithm starts by selecting a single feature randomly and splitting the dataset into two parts—“Training” and “Testing”. It then uses the Testing set to train a decision tree. Once again, it selects a single feature and splits the Training set into two parts, “Training set 1” and “Training set 2”. The algorithm continues this process until all features have been tested on both sets. The process ends when there are no Training data left. For the random forest to make predictions about new cases, it votes for its prediction using all trees in it. To make an accurate prediction, at least 50% of votes should be positive [37]. Decision trees represent a predictive model as a sequence of three layers in which several data variables split the input data space into successively narrower subspaces. Choosing random samples from a dataset is the first step in the random forest algorithm process. This method will then build a decision tree for each sample. Each decision tree’s forecast outcome is then obtained. Subsequently, each anticipated outcome will be put to the vote. Then, the highest voted forecast result as the final prediction result is chosen. The random forest process is simplified in Figure 8.

The random forest algorithm was used because of the following characteristics: compared to the decision tree algorithm, it is more accurate; it offers a practical method for dealing with missing data; it can provide a good forecast without hyper-parameter adjustment; it fixes the overfitting problem of decision trees. A subset of characteristics is chosen randomly at the node’s splitting point in every random forest tree [38].

### 4.2. Detection Model

A pre-infection scenario represents the detection of the spyware during the installation phase; the lifecycle for any spyware system starts with installation. The proposed dataset includes a separate class for spyware installation data which will be used in this model as “Class B”. Post-infection is the second phase within the spyware lifecycle; in this phase, the spyware system is installed and activated with complete operation activity. This means that the spyware fully exercises its espionage functions, which represent collecting data as it has been programmed, sending it to the control center, and receiving other related commands such as deleting specific existing data. Data related to this stage are labeled under “Class C”.

The overall function and workflow of this model can be summarized in the following steps:Step 1: Import the data and determine the detection approach, pre-infection or a post-infection; in this step, you can use online live feed data collected from smartphone traffic or even apply offline data within the model. Import data files using WEKA.Step 2: Train the model. This step involves training the model on a variety of smaller datasets and evaluating them against a smaller testing set. This characteristic is known as K-fold cross-validation. Cross-validation is a statistical technique used to assess the proficiency of machine learning models. A single parameter named k specifies the number of groups into which a given data sample is to be partitioned. When a particular value for k is selected, it may be substituted for k in reference to the model. In this model, k = 10 is used for 10-fold cross-validation. As shown in Figure 9, this stage includes shuffling the dataset randomly and splitting the dataset into ten groups for each unique group. Then, taking the group as a holdout or test dataset, the remaining group as a training dataset, and finally fitting a model on the training set and evaluating it on the test set.


Step 3: Identify the most important features. Your dataset’s attributes (also known as columns or features) are assessed using the attribute evaluation approach (e.g., the class). Several techniques may narrow an extensive data collection to a few valuable characteristics. A significant number of characteristics in a database means that many attributes will not be relevant in the present study. As a result, deleting the dataset’s undesirable features is a critical step in building a solid machine learning model. In this model, we adopted the 18 features listed in Table 10.Step 4: Tuning the model. This step includes searching for the best number of parameters. Primarily, three elements may be adjusted to increase the predictive ability of the model:
○max_features: Generally, increasing the maximum number of features increases the model’s performance since each node must now examine a more significant number of alternatives. However, this is not always true since this reduces the variety of individual trees, which is the unique selling proposition of random forests. However, if you increase the max features, the algorithm’s performance will undoubtedly fall. Therefore, you must strike the appropriate balance and choose the ideal max features.○n_estimators: This is the number of trees you want to plant before calculating the maximum voting or prediction averages. The greater the number of trees, the greater the performance, but the slower the code. It would help to choose the maximum figure your CPU can manage since this will strengthen and stabilize your forecasts.○min_sample_leaf: If you have previously constructed a decision tree, you understand the significance of the minimum sample leaf size. The terminal node of a decision tree is the leaf. A smaller leaf makes the model more susceptible to catching training data noise. In general, a minimum leaf size greater than 50 is preferred. However, it would help if you experimented with various leaf sizes to see which is optimal for your use case [40].Step 5: Evaluate the model. A model’s ability to accurately predict the target based on new and future data should continually be evaluated. The accuracy measure of the ML model must be checked on data for which the target result is already known. This evaluation should be used as a proxy for prediction accuracy on future data since future occurrences have unknown target values.


Figure 10 shows the workflow model in which our novel dataset is being utilized. Multi-class and binary-class classification are accomplished using random forest algorithms. In the algorithm, all processed data are examined to determine whether it is spyware traffic or regular traffic, and the appropriate action is made based on the classifier results. Feature selection is a technique for minimizing the amount of data that goes into your model by removing irrelevant or useless information. Feature selection is the process through which machine learning automatically selects relevant characteristics depending on the sort of issue you are attempting to solve. It has been performed by simply adding and subtracting features without altering them. 

The testing results for this model were within reasonable and acceptable limits, and our goal in this research is to prove the eligibility of the database for use and not to obtain a very high accuracy of results. Therefore, this research did not address any aspects related to feature engineering. 

## 5. Results and Discussion

In this section, we will list and discuss the results we obtained through the use of the random forest algorithm and explain some of the observed phenomena. First, we will define the evaluation metrics we adopted in this experiment. Then, we will list experimental results (accuracy and confusion matrix results). Finally, we will compare our results with other research and discuss the results accordingly. As mentioned before, this result was conducted based on three data classes: Class A, Class B, and Class C.

### 5.1. Evalaution Metrics

There are several measures for testing the effectiveness of the machine learning model. This section aimed to demonstrate the adopted metric measures used for random forest algorithm, including accuracy, true positive rate, F-measure, and confusion matrix. This study evaluated binary- and multi-class classifications, which we adopted in this research. Evaluation metrics for classification algorithm used are based on four main values: TP, FN, FP, and TN [41].
True Positive (TP) represents the number of records labeled “Normal” that are correctly classified as “Normal”.False Negative (FN) represents the number of records labeled “Spyware” but classified as “Normal”.False Positive (FP) represents the number of records labeled “Normal” but classified as “Spyware”.True Negative (TN) represents the number of records labeled “Spyware” that are correctly classified as “Spyware”.

Accuracy is a well-known performance parameter that distinguishes a robust classification model from a poor model when assessing binary classification. Accuracy is computed as in Equation (1).
(1)$$Accuracy=\frac{\left(TP+TN\right)}{\left(TP+TN+FP+FN\right)}\;$$

Precision represents the percentage of all genuinely positive instances despite being anticipated to belong to the positive class. Precision is computed as in Equation (2).
(2)$$Precision=\frac{TP}{TP+FP}$$

Recall, or true positive rate, represents the percentage of instances expected to fall into the positive class. Recall is computed as in Equation (3).
(3)$$TPR=\frac{TP}{TP+FN}\;\;$$

F1-score measures the accuracy. It calculates the harmonic mean of the Precision and Recall. The F-measure’s value is between 0 and 1; when it becomes close to 1, it has achieved the best score, and the results of both Precision and Recall are perfect, and vice versa. F1-score is computed as in Equation (4) [42].
(4)$$F1\;Score=2\times \frac{Precision\times TPR}{Precision+TPR}\;\;\;$$

A classification problem’s predicted outcomes are collected in a confusion matrix. A confusion matrix represents the total values that describe the number of accurate and inaccurate predictions for each class. In simple words, the confusion matrix demonstrates how your classification model produces predictions while being confused. Confusion matrix results will be listed in the binary- and multi-class classification experimental results [43].

### 5.2. Experimental Results

This section will list the experimental results; classification results along with confusion matrix results. The average accuracy was 79% for the binary-class classification and 77% for the multi-class classification. In the multi-class approach, detection accuracy for spyware systems (UMobix, TheWiSPY, MobileSPY, FlexiSPY, and mSPY) was 90%, 83.7%, 69.3%, 69.2%, and 73.4, respectively, and in the binary-class classification, detection accuracy for spyware systems (UMobix, TheWiSPY, MobileSPY, FlexiSPY, and mSPY) was 93.9%, 85.63%, 71%, 72.3%, and 75.96%; respectively.

As shown in Figure 11, the overall average accuracy varies from one system to another. A slight difference is observed between the binary class and the multi-class. The multi-class is generally more accurate because it divides the data into different parts, thus providing more differences between them in the classification process and enabling the classifier to be more accurate. In the binary classification, the result is limited to two cases: (normal data, spyware data).

#### 5.2.1. Detailed Classification Results

In this section, we will list and explain the detailed classification results, carried out through the random forest algorithm, and clarify the meaning of the detailed results table. In general, and to analyze the results of any classification process, four essential criteria must be known, which are the basis for measuring the success of the classification process or not. These criteria can be summarized as follow:Correctly classified instances: This parameter explains the value and percentage of the total number of instances in which the system could classify them correctly. In our case, this value represents the number of instances in which the system successfully classified malicious instances that belong to spyware systems and are directly related to a specific spyware system.Incorrectly classified instances: This parameter determines the number of instances in which the system failed to classify, this means that they are instances that belong to a specific class, but the system incorrectly classifies them as belonging to a different class.Relative absolute error: RAE is a performance metric for prediction models. It is used mainly in machine learning. RAE should not be confused with relative error, an all-encompassing measure of precision or accuracy for devices such as clocks, rulers, and scales.Root relative squared error: RRSP is a performance statistic for predictive models including regression. It is a fundamental indicator that provides the first indication of your model’s performance. Additionally, it is an expansion of the relative squared error (RSE).

Binary classification is a limited classification process that can provide one of two options: normal traffic or spyware traffic. In multi-class classification, we have more specific results; the classifier provides more specific classifying results: normal traffic, spyware installation traffic, and spyware operation traffic. As mentioned before in this research, this dataset can be used in the pre-infection or post-infection process; for this reason, the spyware data are separated into classes: Class B and Class C.

As shown by the overall classification results in Table 11, UMobix achieved the highest true positive rate of 94%, WiSPY achieved an 85% true positive rate, while MobileSPY achieved a 76% TP rate. mSPY achieved a 76% TP rate. Finally, FlexiSPY achieved a 72% TP rate.

#### 5.2.2. Confusion Matrix Analysis Results

The primary issue with classification precision is that it masks the information necessary to measure the success of your classification model. The situation in which this issue is most likely to arise is when your data have many classes. You may get a 90 percent classification accuracy with three or more classes. However, you do not know whether this is because all classes are predicted equally well or if the model is overlooking either one or two classes. This is why a confusion matrix is required, which is a summary of prediction outcomes for a classification task. In simple terms, a confusion matrix is a table that can tell us when our model succeeded in prediction and when the classifier became confused in classification. For FlexiSPY, the classifier model successfully managed to classify 1446 instances out of 3186 instances that belong to FlexiSPY data traffic as normal traffic and was confused in classifying 126 instances of spyware traffic. This represents a percentage of 30% of total malicious instances, which is almost identical to incorrectly classified instances explained in Table 12. In other words, confusion matrix analysis is the proper mathematical model to validate machine learning-based analysis results. Table 13 represents the detailed confusion matrix analysis. As mentioned, we adopted three classes of data (Class A, Class B, and Class C) and two classification methodologies (binary class and multi-class). The binary classification shows the results between two classes Normal (Class A) and Spyware (Class B + Class C); as mentioned before, Class B includes spyware installation data while Class C represents spyware operation data.

A confusion matrix essentially represents results according to four basic parameters, TP (True Positive), TN (True Negative), FP (False Positive), FN (False Negative), and F Score—a weighted average of the true positive rate—which are explained and listed in a number of instances in Table 12 and used to calculate true positive rate and false positive rate, as shown in Table 13.

Finally, from the above results, we can confirm that the novel dataset is mathematically valid and can be relied upon in other advanced research to build different models based on different features, especially the aforementioned feature engineering.

### 5.3. Comparison with Other Works

Comparing scientific research with other research must be adopted on pre-defined criteria. Therefore, we will adopt several benchmarks to compare this research with other research. Targeted spyware systems, deployment approach, analysis technique, analysis algorithm, dataset availability, and analysis results are the primary benchmarks in this comparative analysis.

Conti et al. [15] and Malik et al. [27] confirm research attempts to build detection models for spyware systems. Table 14 includes a scientific comparison between their research and this research based on previously determined criteria. The main drawbacks for Conti et al. [15] include dataset availability, limited targeted spyware systems, and not defining the method that was used to collect the dataset. On the other hand, Malik et al. [27] suggested a detection model based on analyzing DNS requests without analyzing network traffic in general. The main drawbacks in Malik et al. [27] include using generic spyware tools usually embedded within other applications used for essential functions, the absence of a comprehensive monitoring spyware system, and the dataset availability.

After a lengthy and detailed search, it was found that this is the only research that has created a dataset for research purposes for such tools. All other relevant research is based on studying the behavior of these applications in terms of the permissions they request or their programmatic behavior. However, it does not include spyware applications mentioned in this paper.

### 5.4. Discussion

According to the findings that were provided, it is abundantly evident that there is a disparity in the accuracy rate of the suggested model for each spyware system. According to the findings, it is evident that it is possible to rely on the analysis of network traffic data to construct accurate detection models for this kind of application so that they can function during various stages of the life cycle of these systems. 

These spyware applications have a high ability to conceal themselves, as shown by the fact that some of them can hide a significant portion of their digital footprints related to network traffic. One example of this would be the results obtained from the moderate level of accuracy provided by the FlexiSPY and MobileSPY systems. Because of this, further research and development concerning feature engineering are required.

In this particular research, the data were segmented into three distinct classes; each of these classes reflects a distinct phase of the life cycle of the operation of the spyware system, and segmenting it in this manner yielded highly verified findings. This can be seen in the results for the TP rate of Classes B and C.

Because of the use of the unified activity list rather than a spontaneous activity, it is evident from the fundamental structure of the dataset that the overall statistics for the various spyware systems are relatively comparable. The reason for this is the usage of the unified activity list. 

## 6. Conclusions

This paper presented a novel dataset for android-based spyware detection purposes. Initially, an overview of the spyware concept was presented and how it impacts Android-based smartphone privacy. Additionally, an overview of the Android OS security structure is presented better to understand related security threats to users’ privacy. Then, several spyware detection approaches from previous studies were discussed to obtain the knowledge gap and prove the need for a benchmarked dataset.

The presented dataset includes data from the network traffic of the most advanced Android-based spyware tools (FlexiSPY, Mobilespy, mSPY, TheWiSPY, and UMobix); these tools were selected based on relevant, reliable global ratings. A total of 18 features were used during testing on the proposed model. Model testing was conducted using a random forest algorithm to validate the dataset’s usefulness as a benchmarked dataset. Model testing and dataset validation were in two phases: binary and multi-class classifications. It is concluded that the detection model achieved slightly better results in binary classification than in multi-class classification. The model was tested and evaluated for multi-class classification according to the dataset’s main categories represented by normal data, installation data, and operation data categories. The detection model achieved good performance. The average accuracy was 79% for the binary-class classification and 77% for the multi-class classification. In multi-class approach, the detection accuracy for spyware systems (UMobix, TheWiSPY, MobileSPY, FlexiSPY, and mSPY) was 90%, 83.7%, 69.3%, 69.2%, and 73.4, respectively, and in binary-class classification. the detection accuracy for spyware systems (UMobix, TheWiSPY, MobileSPY, FlexiSPY, and mSPY) was 93.9%, 85.63%, 71%, 72.3%, and 75.96%; respectively.

## 7. Future Work

This research opens the pathway for future work in developing and working on spyware detection models. There is a tremendous potential for future research, including various areas of development that can be studied based on this research. Future research contributions can be summarized as follows:Expand the circle of targeted spyware applications, as many spyware programs have not been conducted in similar studies;Analyzing the dataset in this research using other algorithms, as the results of this thesis are based on the random forest algorithm to verify the eligibility and validity of the dataset;Conducting detailed research related to the reverse engineering of these applications and studying its relationship in detail with the results of network traffic analysis.Expand on feature engineering analysis and compare the different results;Develop the dataset to be comprehensive for Apple and Android smartphones.

## Figures and Tables

**Figure 1 sensors-22-05765-f001:**
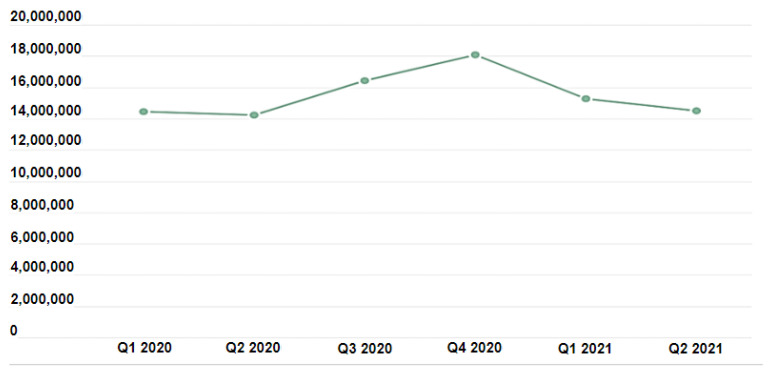
Number of cyberattacks against Kaspersky Android mobile solutions users 2020–2021 [5].

**Figure 2 sensors-22-05765-f002:**
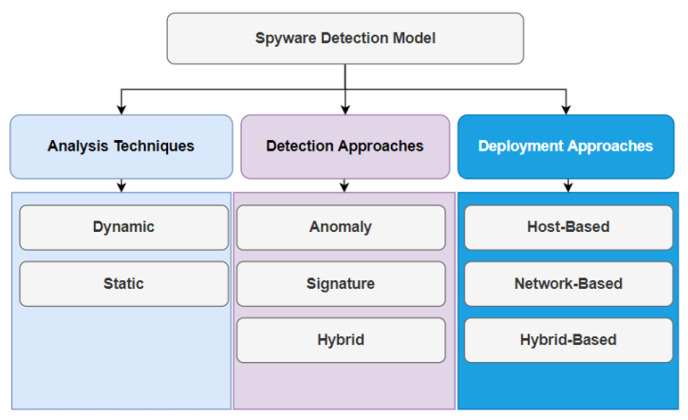
Spyware detection approaches.

**Figure 3 sensors-22-05765-f003:**
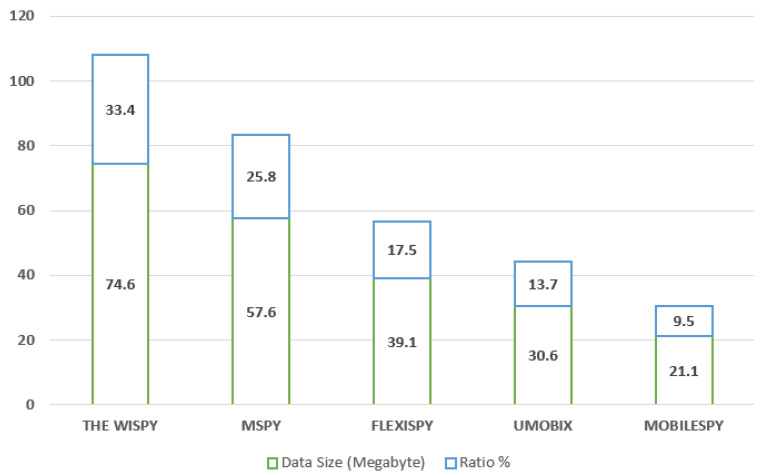
Data distribution and volume.

**Figure 4 sensors-22-05765-f004:**
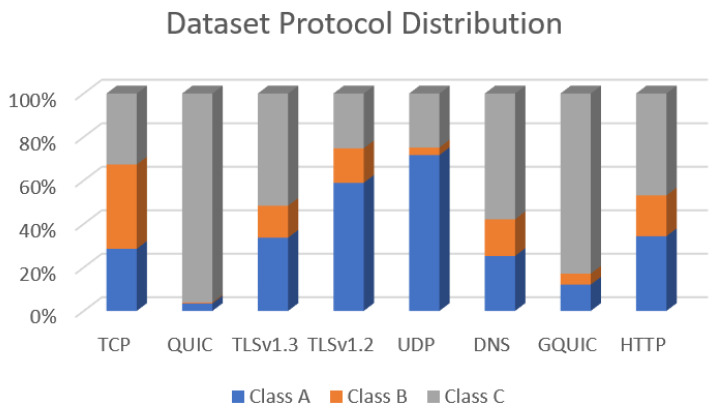
Dataset network traffic protocol distribution.

**Figure 5 sensors-22-05765-f005:**
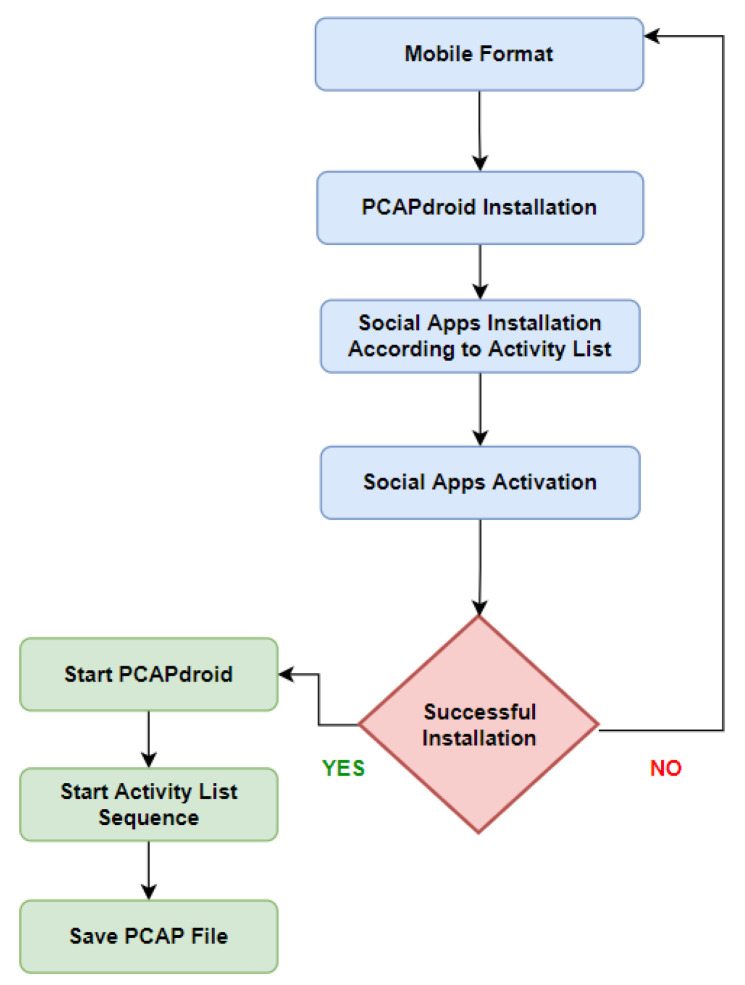
Class A data collection methodology workflow.

**Figure 6 sensors-22-05765-f006:**
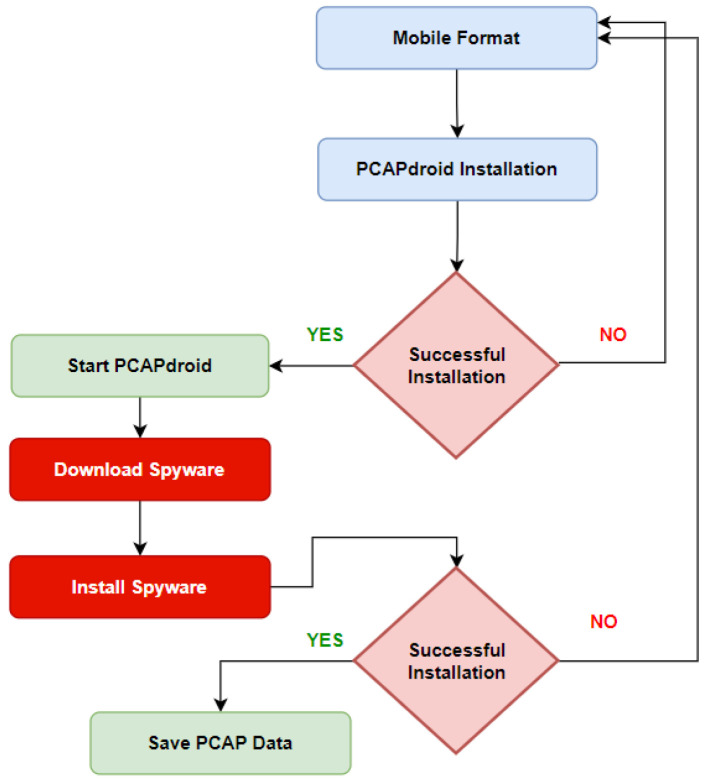
Class B data collection methodology workflow.

**Figure 7 sensors-22-05765-f007:**
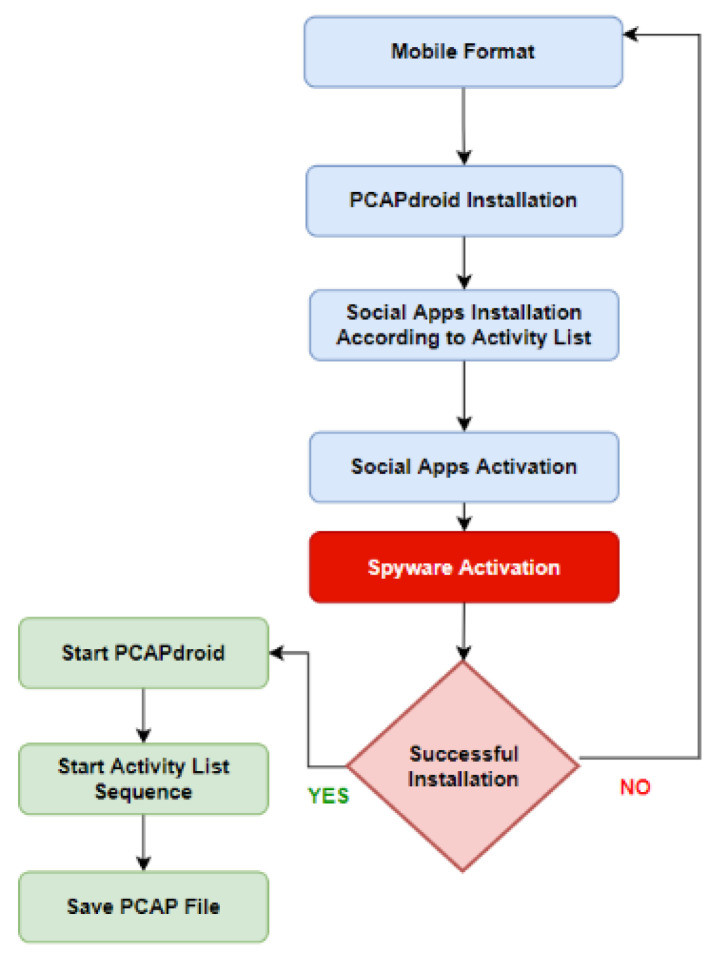
Class B data collection methodology workflow.

**Figure 8 sensors-22-05765-f008:**
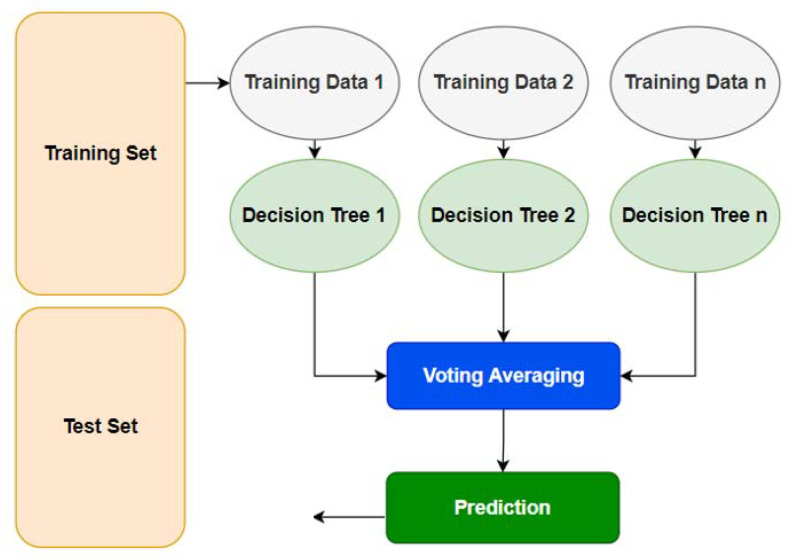
Random forest algorithm for decision tree simplification.

**Figure 9 sensors-22-05765-f009:**
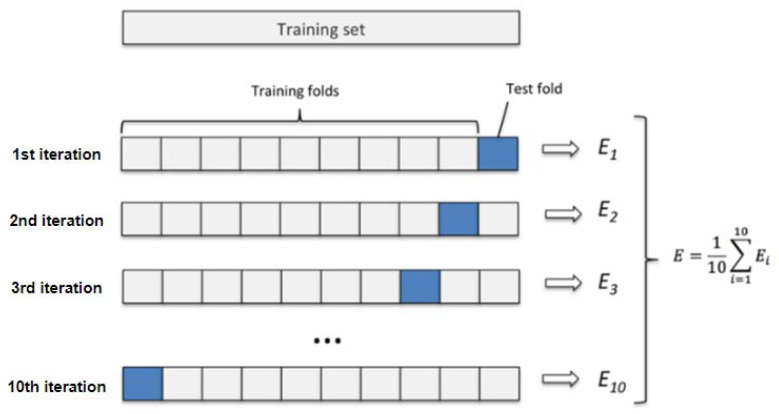
K-fold cross validation process [39].

**Figure 10 sensors-22-05765-f010:**
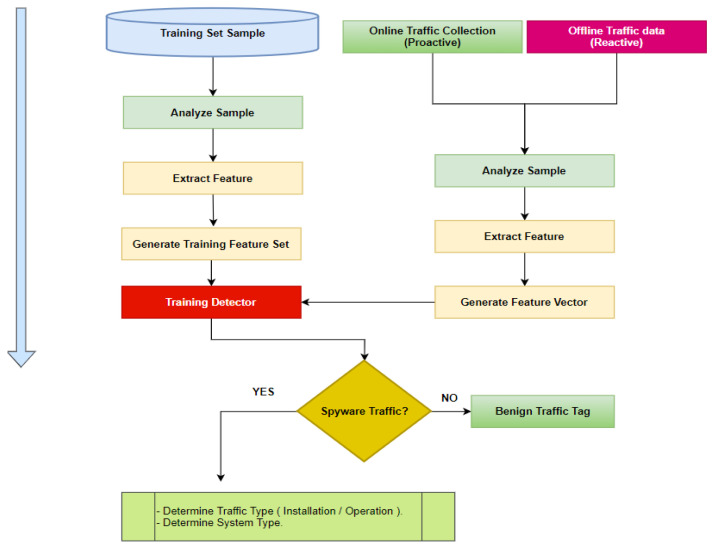
Proposed detection model flow diagram.

**Figure 11 sensors-22-05765-f011:**
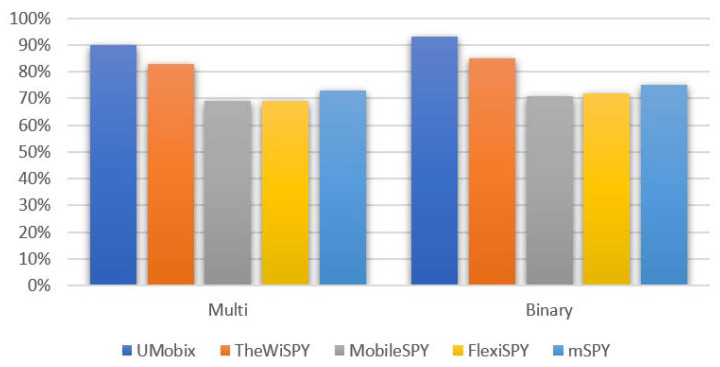
Classification accuracy results.

**Table 1 sensors-22-05765-t001:** Summary of the reviewed state-of-art in this area of study.

Paper	Dataset Type	Permissions/Network Traffic	Data Analysis
Conti et al. [15]	Limited with only one spying system	Network traffic-based	Machine learning-based
Ali-Gombe et al. [17]	Malware only	Permissions-based	Statistical-based
Saad et al. [18]	No dataset	Permissions	Fuzzy logic
Carlsson et al. [19]	No dataset	Permissions-based	Statistical-based
Abualola et al. [20]	Generic dataset	Internal binary code-based	Statistical-based
Pierazzi et al. [21]	Generic dataset	Permissions-based	Machine learning-based
Han et al. [22]	Malware only	Internal binary code-based	Machine learning-based
Kaur et al. [24]	No dataset	Internal binary code-based	Machine learning-based
Vanjire et al. [25]	Malware only	Internal binary code-based	Machine learning-based
Sutter et al. [26]	No dataset	Internal binary code-based	Machine learning-based
Malik et al. [27]	Limited with simple spyware	Network traffic-based	Machine learning-based
Anshul et al. [28]	Malware only	Network traffic-based	Machine learning-based
Taylor et al. [29]	Generic dataset	Network traffic-based	Machine learning-based

**Table 2 sensors-22-05765-t002:** Dataset identification details.

Parameter	Value
Dataset Title	Android spyware
Data Type	PCAP files, CSV files
Data Class	Multivariate
Data Source	Android-based spyware rools
Applications Targeted	FlexiSPY, Mobilespy, mSPY, TheWiSPY, and UMobix
Data Format	PCAP files
Number of Files	24 files
Total Data Size	350 MB
Collection Strategy	Unified activity list
Data Scope	OSI layers 2–7; data link, network, transport, session, presentation, and application
Deployment Approach	Hybrid, host-based, and network-based
Time Constraints	Unified time interval
Number of Classes	Three classes: normal class, installation class, and operation class
File Integrity	MD5
License Type	CC BY 4.0
Data Privacy Compliance	GDPR, PDPC
Dataset Validation Technique	Confusion matrix
Data Collection Tool	PCAPDroid
Data Conversion Tool	CICFlowMeter
Data Preparation Tool	Tamr Unify
Data ML Analyzer Tool	Weka
Other Tools	Dr. Fone Root

**Table 3 sensors-22-05765-t003:** Spyware applications adopted in this research.

System	Cost	Compatibility
mSPY	USD 240	Rooted, non-rooted
uMobix	USD 320	Rooted
MobileSPY	USD 230	Rooted
FlexiSPY	USD 285	Rooted, non-rooted
TheWiSPY	USD 325	Rooted, non-rooted

**Table 4 sensors-22-05765-t004:** High-level summary of the applications’ features.

System	Spying Scope	Platform	Upload	Sniffing
mSPY	- Social media apps - Keylogger - OS activity - Update history - Applications manifest - Phone calls - Microphone	Java, Kotlin	Periodic-b ased with fixed time interval.	Events-based
uMobix	- Social media apps - Keylogger - OS activity - Applications manifest - Phone calls - Microphone	PhoneGap, Java	Adjustable periodic	Adjustable in terms of periodic or event-based.
MobileSPY	- Social media apps - Keylogger - OS activity - Update history - Applications - Phone calls - SIM tracker	React Native, Java	Non-adjustable periodic	Events-based
FlexiSPY	- Social media apps - Keylogger - OS activity - Update history - Applications manifest - Phone calls	Pure Java	Periodic-based with fixed time interval	Events-based
TheWiSPY	- Social media apps - Keylogger - OS activity - Phone calls - Microphone	React Native, Java	Adjustable periodic	Adjustable in terms of periodic or event-based.

**Table 5 sensors-22-05765-t005:** Dataset files list and description.

File Name	System Name	File Size	MD5 Hash	Data Tag
Normal_Traffic.pcap	SmartPhone Normal Traffic	78.81 MB	0151d5fc110f6f7a97ee52be29c99c9a	Normal Traffic
uMobix_Installation.pcap	uMobix	14.37 MB	adab9d323fe85115a8cd8b38fcf45b0a	uMobix Inst.
uMobix_Traffic.pcap	uMobix	16.28 MB	d1a8bbe1e6c0ad85ddb3ae3f0386cf83	uMobix Traffic
TheWiSPY_Installation.pcap	TheWiSPY	53.24 MB	3b45d0ae1f1c9ca9c6b4542a6c956280	TheWiSPY Inst.
TheWISPY_Traffic.pcap	TheWiSPY	21.36 MB	f742fe72b9591a6a66e662eafc991c1b	TheWiSPY Traffic
mSPY Installation Process.pcap	mSPY	11.34 MB	a5c90fbbefeb789fcacce36fd69a830a	mSPY Inst.
Mspy Traffic- Part1.pcap	mSPY	25.94 MB	298d7830454d522524c1fa6e98df9a99	mSPY Traffic
Mspy Traffic- Part2.pcap	mSPY	20.32 MB	4ba6bd67e977126087a542715cf8143e	mSPY Traffic
MobileSpy_Traffic.pcap	MobileSPY	12.76 MB	8d7ec5fef06a896708dc486c6004e9c3	MobileSPY Traffic
Mobilespy_Intallation_01.pcap	MobileSPY	8.41 MB	74200634455d33d5501622213f1ee8d0	MobileSPY Inst
FlexSPY_Traffic.pcap	FlexiSPY	22.32 MB	3baf2d16713f8d94b1ff723061b8de09	FlexiSPY Traffic
FlexiSPY_Installation.pcap	FlexiSPY	16.78 MB	570a6ddfffd72bb4f132823174cade66	FlexiSPY Inst

**Table 6 sensors-22-05765-t006:** Network protocol distribution according to class.

Protocol	Class A	Class B	Class C
TCP	77,679	105,347	88,833
QUIC	1411	196	39,160
TLSv1.3	13,432	5853	20,590
TLSv1.2	22,551	6078	9643
UDP	4020	196	1393
DNS	848	566	1940
GQUIC	161	67	1101
HTTP	33	18	45

**Table 7 sensors-22-05765-t007:** Activity list adopted during data collection.

Activity	Activity Type	Description
Unlocking screen	Operating system security	Unlocking screen, which will trigger an event for spying system to log this event and sent it to control panel, also collecting pin key using keylogger.
Using instant messaging apps (WhatsApp, WeChat, Facebook, QQ, Snapchat, Telegram)	Data exchange, triggering OS APIs related to network infrastructure. Notifications are used by spying systems to sniff such messages.	Simulate a real conversation between two accounts for each app and monitor the spying process to collect exchanged data between the spying client and control panel.
Opening camera and activating voice recording through dashboard panel.	Sensors related, camera and microphone handled under sensors APIs on the Android operating system.	Use the dashboard to open the camera and microphone to start eavesdropping.
Using encrypted end-to-end calls through messaging apps (WhatsApp, Telegram)	Sensors related; notifications related.	Spying systems do rely on notifications to sniff encrypted messages.
File exchange activity	Memory related	Receiving new files from Bluetooth and other communication infrastructure.

**Table 8 sensors-22-05765-t008:** Activity list adopted during data collection.

System	Code Name
mSPY	Update services
uMobix	Play services
MobileSPY	Settings
FlexiSPY	Sync Services
TheWiSPY	System Settings

**Table 9 sensors-22-05765-t009:** Dataset benchmark parameters.

Benchmark Specification	Value
Defined rules	A dataset that can be used to build models capable of detecting spyware on Android efficiently and effectively.
Dataset quality	All training samples generated through real-world process without simulation tools.
Dataset quantity	Almost 14,000 instances collected.
Dataset diversity	The selection of the targeted spyware systems was made after reviewing previous research in this field, which belongs to different companies.
Dataset efficiency	A two-phase data collection adopted for each spyware system to provide more efficiency samples (installation, operation).
Dataset eligibility	Dataset eligibility has been tested using random forest algorithm after analyzing data using CICflowmeter, results has been listed in details and confirmed using machine learning.
Dataset consistency	To guarantee a consistency dataset we adopted a unified activity list that has been applied with respect to time for every spyware listed.
Dataset accessibility	Dataset will be published online.
Dataset documentation	Dataset is fully documented.
Dataset rules testing	A model built using machine learning based on the dataset to detect android spyware, results ranged between 72% and 93% with proper analysis and explanation.

**Table 10 sensors-22-05765-t010:** Network traffic features set.

Feature Name	Description
Src Port	Packet source port
Dst Port	Packet destination Port
Protocol	Packet protocol
Flow duration	Duration of the flow in microseconds
total Fwd Packet	Total packets in the forward direction
total Bwd packets	Total packets in the backward direction
total Length of Fwd Packet	Total size of packet in forward direction
total Length of Bwd Packet	Total size of packet in backward direction
Fwd Packet Length Min	Minimum size of packet in forward direction
Fwd Packet Length Max	Maximum size of packet in forward direction
Fwd Packet Length Mean	Mean size of packet in forward direction
Fwd Packet Length Std	Standard deviation of packet size in forward direction
Bwd Packet Length Min	Minimum size of packet in backward direction
Bwd Packet Length Max	Maximum size of packet in backward direction
Bwd Packet Length Mean	Mean size of packet in backward direction
Bwd Packet Length Std	Standard deviation of packet size in backward direction
Flow Bytes/s	Number of flow bytes per second
Flow Packets/s	Number of flow packets per second

**Table 11 sensors-22-05765-t011:** Classification results.

Parameter	FlexiSPY		MobileSPY		mSPY		TheWiSPY		UMobix	
	Binary	Multi	Binary	Multi	Binary	Multi	Binary	Multi	Binary	Multi
Total Instances	3186		3169		3050		2227		2298	
Correctly Classified	2306	2207	2253	2199	2239	2239	1907	1865	2160	2096
Incorrectly Classified	880	979	916	970	811	811	320	362	138	229
Correct Percent	72.30%	69.20%	71%	69.30%	73.40%	73.40%	85.60%	83%	93.90%	90%
Incorrec Percent	27.60%	30.70%	28.90%	30.60%	26.50%	26.50%	14.30%	16.20%	6%	9.90%
Relative absolute error	70.20%	71.40%	74.20%	75.40%	65%	67.30%	66.10%	70.20%	36.10%	45.90%
Root relative squared error	85.40%	86.20%	87.50%	88.10%	81%	82.60%	81%	84.10%	54.40%	64.30%

**Table 12 sensors-22-05765-t012:** Multi-class confusion matrix results.

T*	FlexiSPY			MobileSPY			mSPY			TheWiSPY			UMobix		
	3186			3169			3050			2227			2298		
	A	B	C	A	B	C	A	B	C	A	B	C	A	B	C
Class A*	1446	282	32	1419	330	11	1503	235	22	1697	42	21	1723	32	5
Class B*	432	676	35	512	765	9	382	658	23	165	136	14	88	321	20
Class C*	126	72	85	71	37	15	91	58	78	102	18	32	24	60	25

T*—Total Instances. Class A*—Normal traffic. Class B*—Spy application operation traffic. Class C*—Spy application installation traffic.

**Table 13 sensors-22-05765-t013:** Detailed accuracy results.

	FlexiSPY		MobileSPY		mSPY		TheWiSPY		UMobix	
	Normal	Malicious	Normal	Malicious	Normal	Malicious	Normal	Malicious	Normal	Malicious
Total Instances	3186		3169		3050		2227		2298	
*TP Rate	0.794	0.637	0.792	0.61	0.826	0.699	0.955	0.486	0.975	0.825
*FP Rate	0.363	0.206	0.39	0.208	0.331	0.174	0.514	0.045	0.175	0.025
F-Measure	0.761	0.674	0.753	0.652	0.799	0.702	0.913	0.587	0.961	0.865

*TP—True Positive. *FP—False Positive.

**Table 14 sensors-22-05765-t014:** Literature comparison list.

Paper	Spyware Systems	Deployment Approach	Analysis Technique	Analysis Algorithm	Dataset Availability	Results
Conti et al. [15]	Cerberus, mSPY, TruthSPY	Network-based	Dynamic technique	RF, k-NN	Not available	RF 85%, k-NN 65%, and 47% for LR
Malik et al. [27]	Generic embedded spyware tools not monitoring systems	Network-based	Static technique	RF	Not available	63%
This research	UMobix, TheWiSPY, MobileSPY, FlexiSPY, and mSPY	Hybrid approach	Hybrid approach	RF	Available under CC BY 4.0	79% for the binary-class classification and 77% for the multi-class classification

## Data Availability

The dataset presented in this study are available in the following URL: https://majdiqabalin.com/datasets/AndroidSDS2022 (accessed on 7 June 2022).

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
