# Peer review of "Android Spyware Detection Using Machine Learning: A Novel Dataset"

_sensors, 2022, doi:10.3390/s22155765_

Round 1

Reviewer 1 Report

This paper should be straight rejected. The novelty is quite low and the presentation is poor. The manuscript contains a laundry list of descriptions for quite a long number of pages. The actual proposed model for spyware detection and classification comes quite later and then, the overall contribution is no more than of a conference level. This is not up to the mark for a journal with SCIE indexing. The case study is shallow and overall, the contribution to the body of knowledge is quite low.

Author Response

Dear Reviewer, 

First of all, I want to thank you for all the comments you made on this paper.

The paper has been rewritten, taking into account the following points:

- The abstract and the conclusion have been completely rewritten, and the results have been fully stated, consistent with the results mentioned in the abstract and the results section.

- The introduction has been rewritten accurately without repeating any content; research aims and contributions were mentioned.

- Within the dataset structure, a distribution table for network packets is listed that includes protocol details.

- The Whole paper structure has been changed according to these outlines:

Abstract.
1- Introduction.
2- Background & Related Works.
3- Dataset
    - Dataset Identification.
    - Dataset Structure.
    - Data Collection Methodology.
        - Class A Collection Methodology
        - Class B Collection Methodology
        - Class C Collection Methodology
    -Dataset Benchmark 

4- Proposed Model

    - Baseline Algorithm.
    - Detection Model
5- Results & Discussion

    - Evalaution Metrics
    - Experimental Results
        - Detailed Classification Results
        - Confusion Matrix Analysis Results
    - Comparison with other works
    - Results Discussion
6- Conclusion
7- Feature Work

All the comments mentioned by you were taken in detail and processed in addition to the above.

A full language review was done by a native speaker.

Once again, we thank you for reviewing this paper. We apologize for the delay in making the changes, as major changes have been made.

BR

Reviewer 2 Report

This paper presents a method to collect dataset based on network traffic monitoring and uses it to train the machine learning algorithm random forest to detect android spyware.

It is an interesting paper. However, it is not well presented. There are errors, typos, inconsistencies. Some well-known or not closely related contents are unnecessarily presented in detail. Moreover, there is no comparison (such as accuracy, efficiency, etc.) of the proposed method to other existing works.

- Line 297: … Each spy system data file, data colleting methods, and tools that have been adopted. Will also explain data verification process. In addition to a superficial reverse engineering analysis for these applications. …

Grammarly incorrect

Line 354: … Root access is likened to jailbreaking smartphones running Apple's iOS operating system. …

Since this section is about dataset, it is better to focus on dataset rather than unrelated topic jailbreaking.

Line 372: … Its known that the method of each system transferring data differs …

It’s?

Line 454: … Our novel dataset includes detailed information about commonly used network features that are essential in in detection process, network traffic features set are explained in Table5 …

Delete duplicate in

Line 473: … Most of the systems provide a dedicated dashboard that can be used to check connection status as in Figure5 for mSPY system ...

It is not clear about the relation of this paragraph and the section topic Data Collection Methodology (e.g. does PCAPdroid need the dashboard to collect data?).

Line 475: … One the process of collecting application data installation phase is done, and the integrity of collected PCAPdroid files is verified ...

Once?

Line 481: … The selection of these activities was based on the global “Statistica” network ...

Statistics?

Lines 484 - 490: … Instant messaging is a type of online chat that lets you send and receive text messages in real-time ...

There is no need to explain well-known instant message in detail.

Lines 518 - 523: … Dataset copyrights must be available to the public …

Since copyright, license, copyleft are not closely related contents, there is no need to discuss them in detail.

Line 566: … It then uses the Testing set to train a decision tree ...

Testing set or training set?

Line 653: In this section we to prove the scientific eligibility of this dataset.

Delete to?

Abstract: … Results ranged between 65% and 92% …

Figure 8: … between 69% and 90% …

Conclusion: … results ranged from 72% to 93% …

The accuracy results are not consistent. In addition, what are the accuracy results of other works like those in table 1? It should compare the proposed method (such as accuracy, efficiency, etc.) to other existing works.

Author Response

(The authors gave the same response as above.)

Reviewer 3 Report

This is a very interesting and important article. Using APKTOOL for reverse engineering is reasonable but steps to do so should be mentioned a little bit for the novice readers to follow. The following must be addressed and improved before publication:

1) The writing is quite bad as it is quite colloquial. Sometimes the tone used is quite conversational. This must be eradicated as this research is assumed to be a highly professional and academic article. Besides, there are lots of grammatical mistakes, e.g. line #24, 78, 298, 306, 475, etc...

2) The abstract writing is quite redundant. Some contents have been repeated immediately.

3) There is no reference for "Kaspersky" done.

4) The first paragraph inside para 3.1 must be re-written.

5) The "Introduction" contains too many things and the sequence is not logical. The authors should brush up and then create a para for "Research Methodology" and also mention the contribution at the Conclusion.

6) There is a major flaw in this article in the Proposed Model section. Although the authors have mentioned about the advantages of Random Forests (RF), they should also mention the other similar methods and then compare them and then arrive at using RF.

7) From line #72 to 75, the authors have defined the terms by themselves. It is better to quote the definitions from the references and then re-interpret them suitably inside the article.

Overall speaking, it is a good article which is worthwhile to get published after the above points have been improved.

Author Response

(The authors gave the same response as above.)

Round 2

Reviewer 1 Report

The revised paper reads even more ridiculous. The presentation became so clumsy with all those correction tracking underlines! Again, to summarize it, this is mainly text revision without really advancing anything. If the claim of contributing a dataset is made with some technique, neither one got justice in the work. On top of that, the authors seemed not to even care about grammar or language or tried to hurriedly prepare another version to push it through the system! The section 7 for instance is titled,

"7. Feature Work"

Feature work? Or, Future Work? You are talking about future work. Aren't you?

Sloppiness in writing, incongruent discussion, lack of proper flow of text and apparently disjoint items put together and poor contribution, if any, overall describe the paper.

Strong Reject. I am not willing to review it any further.

Author Response

Dear Reviewer,

We appreciate and respect your point of view on this research, but we believe that the language of criticism should be scientific, far from abuse like the use of  "ridiculous paper."

Regarding Feature work, corrected to "Future work." 

BR

Reviewer 2 Report

The revision addressed some of my concerns. However, the newly added subsection "Comparison with other works" is not very well written, better to improve the quality of presentation. Moreover, it compares this work to Conti's work rather than other works listed in table 1 because they are not close to this work. But it seems that Malik's work has a very similar configuration to Conti's work in table 1, should explain why it cannot be compared.

Author Response

Dear Sir,

First, we would like to thank you for the observations that enriched this research.

Point 1: Comparison with other works

Response 1: This paragraph has been completely reformulated. A table has been added that includes a complete comparison of this research with another research (Conti et al. [15], Malik et al. [27]). Predefined comparison parameters listed within the table include Targeted spyware systems, deployment approach, analysis technique, analysis algorithm, dataset availability, and analysis results.

Again, Thank you for all of your rich comments.